# Tree Diversity and Its Ecological Importance Value in Silvopastoral Systems: A Study along Elevational Gradients in the Sumaco Biosphere Reserve, Ecuadorian Amazon

Bolier Torres [1,2,3], Robinson J. Herrera-Feijoo [3,4], Alexandra Torres-Navarrete [5], Carlos Bravo [6] and Antón García [2,*]

1. Facultad de Ciencias de la Vida, Universidad Estatal Amazónica (UEA), Puyo 160101, Ecuador; btorres@uea.edu.ec
2. Animal Science Department, University of Cordoba, Rabanales University Campus, 14071 Cordoba, Spain
3. Ochroma Consulting & Services, Tena 150150, Ecuador; rherreraf2@uteq.edu.ec
4. Facultad de Ciencias Agrarias y Forestales, Universidad Técnica Estatal de Quevedo (UTEQ), Quevedo Av. Quito km, 1 1/2 Vía a Santo Domingo de los Tsáchilas, Quevedo 120550, Ecuador
5. Facultad de Ciencias Jurídicas, Sociales y de la Educación, Universidad Técnica de Babahoyo, Extensión Quevedo (UTB), Km 3 1/2 Vía a Valencia, Quevedo 120303, Ecuador; rtorresm@utb.edu.ec
6. Facultad de Ciencia de la Tierra, Universidad Estatal Amazónica (UEA), Puyo 160101, Ecuador; cbravo@uea.edu.ec
* Correspondence: pa1gamaa@uco.es

**Abstract:** This study analyzes tree diversity and its ecological importance value in silvopastoral systems in the Sumaco Biosphere Reserve (SBR), Ecuador, along an altitudinal gradient of 400–2000 masl. Twenty-six plots distributed into low (400–700 masl), medium (701–1600 masl) and high (1601–2000 masl) zones were used. The Shannon index and the importance value index (IVI), based on abundance, dominance and relative frequency, were estimated. The results highlight that in pastures with dispersed trees, the richness of trees decreases with increasing altitude in the elevational gradient; they also show a higher tree density at lower altitudes in contrast to the Andean–Amazonian primary forests. The lower and middle zones showed higher diversity, linked to regeneration and the presence of nearby forests. Species of high commercial value, such as *Cedrela odorata* and *Jacaranda copaia*, were common, reflecting knowledge of the local timber market. In the lower and middle zones, the 10 most important species accounted for more than 70% of the trees, with up to 96% in the upper zone. A total of 51 taxa (including 42 species and nine taxa at the rank of genus) were identified, which were mostly native; 64.7% classified by the IUCN as least-concern (LC) species, 31.4% as not evaluated (NE) species and 3.9% as vulnerable (VU) species, specifically highlighting *Cedrela odorata* and *Cedrela montana*. The study concludes with policy recommendations related to the importance of trees in silvopastoral systems for the conservation of species and the livelihoods of local communities, highlighting the need for responsible management of Amazonian pasturelands.

**Keywords:** trees in grasslands; diversity indices; endangered species; Ecuadorian Amazon

## 1. Introduction

Tree diversity is a key element in ecological resilience and the provision of multiple ecosystem services in traditional silvopastoral systems [1–3]. These systems, which integrate dispersed trees in pastures, are being recognized for their environmental [4–6] and economic benefits [7–9], highlighting the importance of tree diversity, which motivates an intense analysis along the Andean–Amazonian altitudinal gradient [7,10].

The presence of trees in pastures favors the improvement of biodiversity [11] and soil quality [12] while enhancing carbon sequestration and the provision of other key ecosystem services [7,13]. The capture and storage of atmospheric carbon is essential for climate change mitigation, and trees on pastures contribute significantly to this process [4,14].

Several studies have demonstrated that trees in silvopastoral systems act as carbon sinks and store large amounts of carbon in the tree biomass and in the soil [13,15], which not only reduces the concentration of carbon dioxide in the atmosphere but also promotes the resilience of the systems to extreme climatic events by maintaining the stability of the microclimate [16,17].

On the other hand, trees generate a direct economic value as a consequence of the production of fruits and wood of commercial interest [18,19]. The presence of fruit trees in silvopastoral systems not only diversifies the diet of livestock but can also generate additional income for farmers [20]. Likewise, timber trees, such as those destined for the construction industry or furniture manufacturing, offer long-term sustainable economic opportunities [21]. This multifunctionality of trees in silvopastoral systems underlines their value as key elements for resilience and sustainability in both environmental and economic terms [2,13], emphasizing the need for a conscious management of tree diversity in forests [22].

According to Montagnini and del Fierro [23], the presence of tree species in silvopastoral systems can promote nutrient retention, reduce erosion and improve soil quality. The positive effect on soil health has direct implications for livestock productivity and the long-term sustainability of these systems. Therefore, the development of an exploratory study about tree diversity and its ecological importance value in silvopastoral systems is a priority step to optimize natural resource management and improve the resilience of agro-livestock activities in the current context of environmental and climatic challenges [1,2,24].

In this context, this study, conducted in an area widely recognized as a hotspot of biodiversity and endemism [25–27], the Ecuadorian Amazon Region (RAE), has the following objectives: (a) to know the tree diversity in grasslands with scattered trees; (b) to determine the abundance of tree species and their value of ecological importance; and (c) to analyze the density of trees and their conservation status in Ecuador according to the International Union for Conservation of Nature (IUCN) in silvopastoral systems along the altitudinal gradient of the Sumaco Biosphere Reserve (SBR) in the Ecuadorian Amazon.

## 2. Materials and Methods

### 2.1. Study Area

The study was conducted among households engaged in livestock-based livelihood strategies [28] in an altitudinal gradient within the Sumaco Biosphere Reserve (SBR). The SBR is about one million ha in size [29], and according to the last multitemporal assessment conducted in 2013, it consisted of about 53% primary forest, 28% secondary forest and 9% grassland (81,693 ha) [29]. Three Amazonian cantons located within the SBR were selected for this study: (a) Arosemena Tola (low zone from 400 to 700 masl), (b) Archidona (middle zone from 701 to 1600 masl) and (c) Quijos (Amazon high zone from 1601 to 2000 masl) (Figure 1). The whole study area is part of one of the world's biodiversity hotspots (the Western Amazon Highlands) [27]. The predominant bioclimatic conditions vary along the altitudinal gradient studied, with a mean annual temperature of 35.67 °C and annual precipitation of 5209 mm in the low zone, 33.65 °C and 4728 mm in the middle zone and 26.70 °C and 2205 mm in the high zone [7].

### 2.2. Sampling and Data Collection

The selection criteria for farm selection were a pasture area ≥0.5 ha with at least one pasture plot with scattered trees and canopy cover ≥10% determined with a spherical densitometer [7,30]. Thus, 26 circular temporary plots of 2826 $m^2$ were installed in pasture with scattered trees, distributed among the elevational gradients as follows: twelve, eight and six plots in the lower, middle and high zones, respectively (Figure 2 and Table 1).

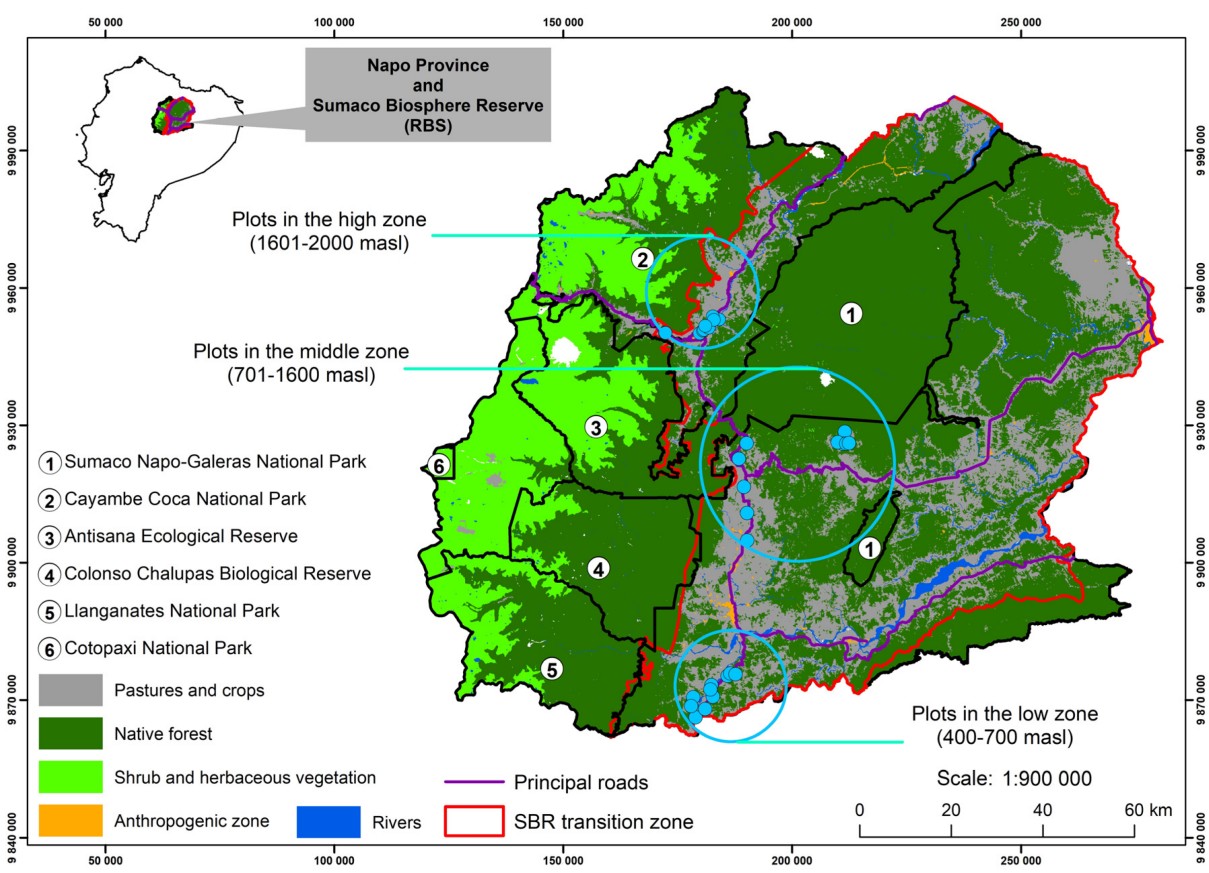

**Figure 1.** Study area location and temporary plots established along the altitudinal gradient.

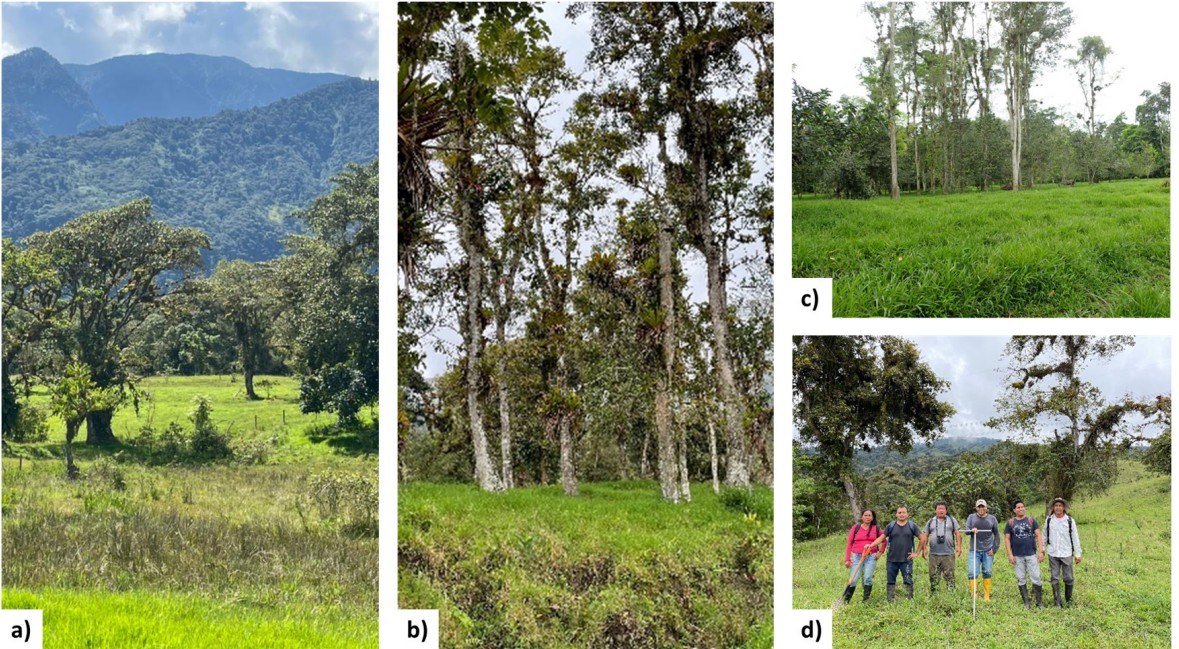

**Figure 2.** Grassland system with scattered trees: (**a**) Amazonian high zone, (**b**) middle zone and (**c**) low zone; (**d**) team in the Amazonian high zone.

**Table 1.** Plot characteristics of livestock producers along the altitudinal gradient, Sumaco Biosphere Reserve, Napo Province, Ecuadorian Amazon.

| Variable | Amazon Altitudinal Gradient (Zone) | | |
| :---: | :---: | :---: | :---: |
| | **Low** | **Middle** | **High** |
| Elevation range (masl) | 400–700 | 701–1600 | 1601–2000 |
| Number of plots | 12 | 8 | 6 |
| Total area sampled (ha) | 3.39 | 2.2 | 1.69 |
| Average elevation (masl) | 543.1 | 1114.1 | 1778.0 |
| Year of settlement | 1975 | 1984 | 1952 |
| Ethnicity of head of household (% Kichwa) | 0.0 | 56.1 | 0.0 |

*2.3. Data Analysis*

The Shannon diversity index (*H*) of each species was determined to consider both the abundance and the variety of species present in each zone along the studied gradient [31,32].

$$H = -\sum_{i=1}^{S}[(P_i) * ln(P_i)], \tag{1}$$

where *S* is the number of species present, *ln* is the natural logarithm and *Pi* is the proportion of individuals found of species *i*; it is calculated by the ratio (*ni/N*), where *ni* is the number of individuals of species *i* and *N* is the total number of individuals. However, considering that Shannon's index is a measure of the entropy present in a system but not diversity per se [33], we use the exponential Shannon index ($H_{exp\ Shannon}$) described by Jost [34] that allows us to describe the levels of diversity among the altitudinal gradients:

$$H_{exp\ Shannon} = exp\left(-\sum_{i=1}^{S}[(P_i) * ln(P_i)]\right), \tag{2}$$

The Shannon equitability index *E* is calculated as

$$E = \left(-\frac{H}{\sum_{i=1}^{S}\left(\frac{1}{S} * ln\left(\frac{1}{S}\right)\right)}\right) * 100, \tag{3}$$

where the denominator is the maximal possible *H* and *E* ranges from 0 to 100 and reflects the share of the actual diversification in relation to the maximal possible diversity.

Simpson's diversity index (D) was also obtained to estimate the importance of taxa with high value [35]:

$$D = \Sigma P_i^2, \tag{4}$$

where *Pi* is the proportion of species *i* in the community (*ni/N*), *ni* is the number of individuals of species *i* and *N* is the total number of individuals. We also used the inverse of the Simpson index ($D_{inv\ Simpson}$), using the following formula [34]:

$$D_{inv\ Simpson} = 1/\Sigma P_i^2 \tag{5}$$

EstimateS v.9.1.0 was used to calculate species richness, dominance, density of individuals and rarefaction curves [36] to statistically represent the accumulation of species in relation to the number of samples, which is useful in several sampling approaches [37].

The importance value index (IVI) was calculated according to Curtis and Macintosh [38] for each species:

$$[IVI = \text{relative abundance (R.A.)} + \text{relative dominance (R.D.)} + \text{relative frequency (R.F.)}], \tag{6}$$

where:

- Relative abundance R.A. = percentage of number of individuals ha$^{-1}$;
- Relative dominance R.D. = percentage of basal area (m$^2$ ha$^{-1}$);

- Relative frequency R.F. = percentage of plots in which a species is present.

SPSS 22.0 for Windows software was used to perform the statistical analyses. Each sample collected was considered a priori as a discrete group. Prior to the statistical analyses, the normality of the data distribution was evaluated using the Kolmogorov–Smirnov test (with Lilliefors correction). For those variables that did not show a normal distribution, the Bartlett test was applied to assess whether the data had equal variances. Quantitative variables (original and adjusted) were analyzed by means of a one-way analysis of variance (ANOVA), while qualitative variables were compared with the Kruskal–Wallis test [35].

In addition, using the Global Wood Density Database [39,40], wood density values (g/cm$^3$) were recorded by species, a critical parameter for assessing silvopastoral systems. This indicator reflects important ecological and functional properties, such as carbon sequestration, which contributes to climate change mitigation, and mechanical strength, which is essential for determining the potential for sustainable wood use [41,42]. With these considerations in mind, the density values for the species with the highest density were taken from the results presented by Ketterings et al. [43].

On the other hand, regarding the uses of the analyzed species, a verification of the main uses (edible, medicinal, handicraft, material) was carried out in the useful plants of Ecuador published by De la Torre et al. [44]. Finally, with respect to the IUCN categories, the "iucn_summary" function of the taxize package [45], developed for the R programming language environment, was used.

## 3. Results and Discussion

### 3.1. Richness, Diversity Index and Structural Parameters of Silvopastoral System

The results of the 26 temporary plots of 2826 m$^2$ show a significant variation in species and family richness along the altitudinal gradient in the SBR. In the lower zone (lower altitude), the highest species richness was recorded, with an average of 10.17 ± 3.21 species per hectare, followed by the middle zone with 6.63 ± 2.72 and the Amazon high zone with 5.53 ± 2.51 (Table 2). These differences are highly significant ($p < 0.01$) and show a decrease in species and family diversity with increasing altitude. This pattern is contrary to the results of species richness found in primary forests in the same area of SBR by Torres et al. [7] in the Ecuadorian Amazon, as well as in a protected primary forest in the Tumbesian dry forest ecoregion [46], where the number of tree species increased with increasing altitude. This variation trend in primary forests may be due to the fact that species distribution patterns are the result of multiple ecological processes [47], influenced by geographic differences and environmental factors such as climate and soil [48], but on the other hand, the evidence of decreasing species richness with increasing altitude in anthropic systems, such as the dispersed trees in these pasture systems, could be the result of the establishment and management of production systems generated by populations of mestizo settlers who came at different times and with different cultural backgrounds and used the ecosystem for different purposes, according to Torres et al. [10]. The first livestock settlements in the SBR occurred in the Amazon highland zone about 70 years ago, then in the lower zone about 45 years ago and, finally, in the middle zone about 35 years ago, which is in agreement with Lei and Zhouping's reports [49] that suggest different stages of succession showed different species composition in natural pastures studied in China.

The exponential Shannon index, which measures species diversity, shows no significant differences between the three altitudinal zones, indicating a similar relative distribution of species in all of them. Similarly, the inverse Simpson's index and Equity index do not show significant differences. Regarding tree density, it is observed that the low zone shows the highest density with an average of 193 ± 97.23 trees per hectare, followed by the high zone with 101 ± 41.54 and the medium zone with 83.25 ± 38.33. These differences are significant ($p < 0.01$) between the low zone compared to the medium and high zones. In contrast, basal area (m$^2$), average diameter at breast height (DBH) and maximum DBH did not show significant differences between the three altitudinal zones studied.

**Table 2.** Averages and standard deviations of floristic composition, diversity index and structural parameters in plots (2826 m$^2$) along the altitudinal gradient in Napo, Sumaco Biosphere Reserve, Ecuadorian Amazon.

| Variable | Amazon Altitudinal Gradient (Zone) | | | Average | *p*-Value [1] |
|---|---|---|---|---|---|
| | **Low** <br> *n* = 12 | **Middle** <br> *n* = 8 | **High** <br> *n* = 6 | | |
| Richness (species) | 10.17 ± 3.21 [a] | 6.63 ± 2.72 [ab] | 5.53 ± 2.51 [b] | 7.96 ± 3.49 | *** |
| Richness (family) | 8.50 ± 2.23 [a] | 6.13 ± 2.58 [ab] | 4.67 ± 1.63 [b] | 6.88 ± 2.68 | *** |
| Exponential Shannon ($H_{exp\ Shannon}$) | 5.62 ± 2.42 | 5.04 ± 2.83 | 3.77 ± 1.42 | 5.02 ± 2.40 | n/s |
| Inverse Simpson ($D_{inv\ Simpson}$) | 4.06 ± 1.82 | 4.18 ± 2.67 | 3.09 ± 1.06 | 3.87 ± 1.97 | n/s |
| Equity (*E*) | 0.57 ± 0.19 | 0.73 ± 0.09 | 0.71 ± 0.12 | 0.65 ± 0.16 | n/s |
| Tree density (ha$^{-1}$) | 193 ± 97.23 [a] | 83.25 ± 38.33 [b] | 101 ± 41.54 [b] | 138 ± 87.50 | *** |
| Basal area m$^2$ (ha$^{-1}$) | 8.67 ± 4.23 | 4.19 ± 3.65 | 6.03 ± 4.97 | 6.68 ± 4.53 | n/s |
| Average DBH (cm) | 20.32 ± 5.39 | 22.37 ± 11.82 | 22.85 ± 12.69 | 21.53 ± 9.24 | n/s |
| Maximum DBH (cm) | 27.78 | 40.67 | 41.54 | 41.54 | - |

[1] *p*-Value: *** $p < 0.01$; n/s = nonsignificant. Letters in superscript denote significant differences between altitudinal gradients.

Tree species rarefaction curves indicate lower tree species richness in grasslands in the high zone compared to the low and middle zones, even if we analyze it with the minimum number of six plots in the three zones. These differences in species richness follow the same pattern both in the analysis by number of plots and by number of individuals (Figure 3).

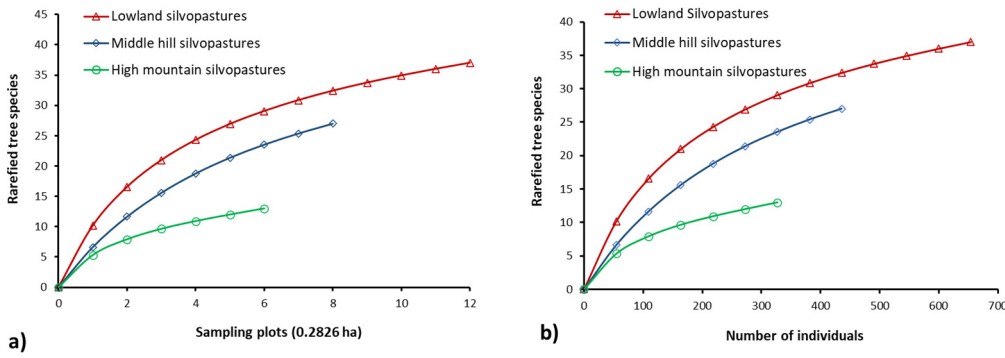

**Figure 3.** Rarefaction curves: (**a**) tree species richness based on temporary plots of 2826 m$^2$ and (**b**) tree species richness based on individuals (trees with DBH ≥10 cm).

Figure 4 shows the distribution of trees at the diameter class level (DBH ≥ 10 cm) in the three altitudinal gradients. In the lower zone, there was a greater presence of trees with diameters less than 30 cm (DBH), with an average of 155 individuals/ha$^{-1}$ in this range. In addition, 31 individuals were recorded/ha$^{-1}$ with diameters between 30 and 50 cm (DBH), while only 6 individuals were recorded/ha$^{-1}$ with diameters greater than 50 cm (DBH). In contrast, in the middle zone, trees with a DBH of less than 30 cm predominated, with an average of 68 individuals/ha$^{-1}$. There were an average of 11 individuals/ha$^{-1}$ with a DBH between 30 and 50 cm, and there were 39 individuals/ha$^{-1}$ with an average DBH of 50 cm. Finally, in the high zone, 87 individuals were identified/ha$^{-1}$ with a DBH less than 30 cm. An average of 8 individuals/ha$^{-1}$ had a DBH between 30 and 50 cm, and an average of 6 individuals had a DBH greater than 50 cm/ha$^{-1}$. The causes of the differences in tree abundance by diameter class are attributed to knowledge of management and years of establishment, as discussed in Section 3.1.

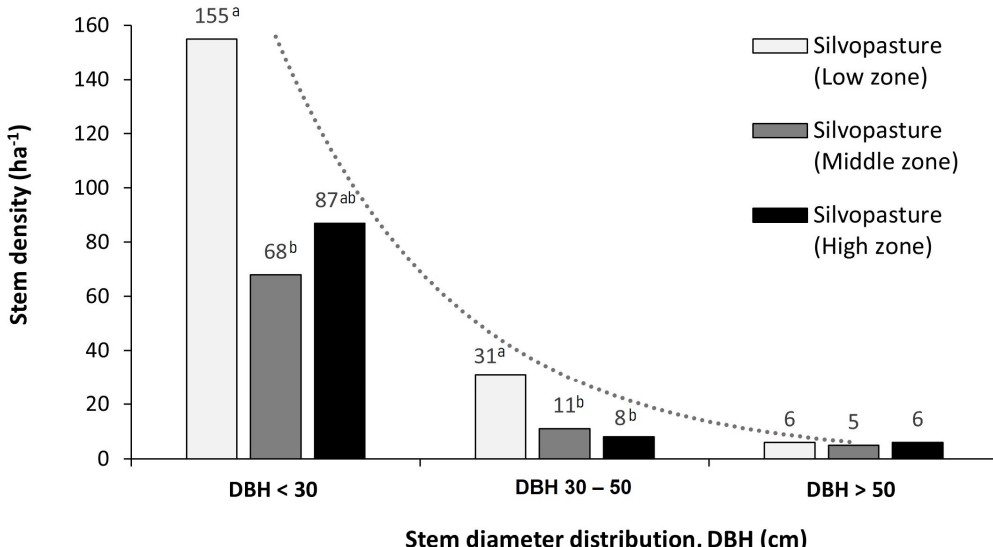

**Figure 4.** Stem diameter distribution (DBH ≥ 10 cm) at three altitude levels in silvopastoral systems. Letters in superscript denote significant differences between altitudinal gradients ($p < 0.01$).

*3.2. Abundance of Tree Species and Ecological Importance Value*

Among the most important shade tree species in the lower zone, *Cordia alliodora* (Cordiaceae) stands out with a high relative abundance (RA) of 27.37% and a relative density (RD) of 16.97%. In the middle zone, *Cordia alliodora* continues to be important, but there is an increase in the dominance of *Jacaranda copaia* (Bignoniaceae) with an RD of 21.12%. In addition, *Piptocoma discolor* (Asteraceae) has a high relative frequency (RF) of 11.32%. In the high zone, *Ficus* sp. (Moraceae) emerges as the most relevant species, with an RA of 5.85% and RD of 53.51%. *Heliocarpus americanus* (Malvaceae) also stands out with an AR of 25.15% and an RD of 21.41% (Table 3).

**Table 3.** List of the most important shade tree species with their relative abundance (RA), relative frequency (RF), relative dominance (RD) and important value indices (IVI) in 26 plots (2826 m$^2$) along the studied gradient in Napo, SBR.

| Family | Species | RA (%) | RF (%) | RD (%) | IVIs |
|---|---|---|---|---|---|
| Silvopasture low zone (400–700 masl) | | | | | |
| Cordiaceae | *Cordia alliodora* | 27.37 | 7.38 | 16.97 | 17.24 |
| Bignoniaceae | *Jacaranda copaia* | 12.84 | 4.92 | 21.12 | 12.96 |
| Myrtaceae | *Psidium guajava* | 17.58 | 8.20 | 3.94 | 9.91 |
| Vochysiaceae | *Vochysia braceliniae* | 4.43 | 4.92 | 10.92 | 6.76 |
| Meliaceae | *Cedrela odorata* | 3.21 | 5.74 | 4.68 | 4.54 |
| Fabaceae | *Piptadenia pteroclada* | 1.53 | 4.92 | 5.44 | 3.96 |
| Myristicaceae | *Virola flexuosa* | 2.60 | 1.64 | 6.99 | 3.74 |
| Lauraceae | *Ocotea* sp. | 1.99 | 5.74 | 3.16 | 3.63 |
| Lauraceae | *Nectandra* sp. | 2.45 | 4.10 | 3.23 | 3.26 |
| Asteraceae | *Piptocoma discolor* | 2.75 | 5.74 | 0.93 | 3.14 |
| Subtotal | | 76.76 | 53.28 | 77.37 | 69.14 |

**Table 3.** *Cont.*

| Family | Species | RA (%) | RF (%) | RD (%) | IVIs |
|---|---|---|---|---|---|
| | | Silvopasture middle zone (701–1600 masl) | | | |
| Cordiaceae | *Cordia alliodora* | 12.23 | 5.66 | 26.39 | 14.76 |
| Asteraceae | *Piptocoma discolor* | 17.02 | 5.66 | 13.21 | 11.97 |
| Meliaceae | *Cedrela odorata* | 15.96 | 7.55 | 4.28 | 9.26 |
| Fabaceae | *Inga* sp. | 5.85 | 11.32 | 6.99 | 8.06 |
| Myrtaceae | *Psidium guajava* | 10.64 | 5.66 | 1.78 | 6.03 |
| Fabaceae | *Inga edulis* | 3.19 | 5.66 | 5.72 | 4.86 |
| Lecythidaceae | *Grias neuberthii* | 1.60 | 5.66 | 6.08 | 4.44 |
| Burseraceae | *Dacryodes peruviana* | 3.19 | 1.89 | 5.92 | 3.66 |
| Urticaceae | *Cecropia membranacea* | 1.60 | 5.66 | 3.10 | 3.45 |
| Lauraceae | *Nectandra* sp. | 5.32 | 3.77 | 1.25 | 3.45 |
| Subtotal | | 76.60 | 58.49 | 74.72 | 69.94 |
| | | Silvopasture high zone (1601–2000 masl) | | | |
| Moraceae | *Ficus* sp. | 5.85 | 12.50 | 53.51 | 23.95 |
| Malvaceae | *Heliocarpus americanus* | 25.15 | 18.75 | 21.41 | 21.77 |
| Myrtaceae | *Psidium guajava* | 38.01 | 9.38 | 6.38 | 17.92 |
| Lauraceae | *Ocotea* sp. | 11.11 | 15.63 | 5.07 | 10.60 |
| Fabaceae | *Inga* sp. | 9.94 | 9.38 | 5.56 | 8.29 |
| Lauraceae | *Nectandra* sp. | 4.68 | 9.38 | 0.99 | 5.01 |
| Rutaceae | *Citrus limon* | 1.75 | 6.25 | 0.33 | 2.78 |
| Meliaceae | *Cedrela montana* | 0.58 | 3.13 | 2.41 | 2.04 |
| Phyllanthaceae | *Hieronyma oblonga* | 0.58 | 3.13 | 1.80 | 1.84 |
| Euphorbiaceae | *Croton lechleri* | 0.58 | 3.13 | 1.55 | 1.75 |
| Subtotal | | 98.25 | 90.63 | 99.00 | 95.96 |

The important value indices (IVIs) reveal the most influential species in each altitudinal zone. In the low zone, *Cordia alliodora*, *Jacaranda copaia* and *Psidium guajava* have high IVIs of 17.24, 12.96 and 9.91, respectively. Meanwhile, in the middle zone, *Cordia alliodora* and *Piptocoma discolor* are dominant with IVIs of 14.76 and 11.97, respectively. Finally, in the high zone, *Ficus* sp. and *Heliocarpus americanus* stand out with IVIs of 23.95 and 21.77, respectively. The IVI made it possible to identify the 10 tree species with the greatest ecological importance in the pasture systems with dispersed trees along the altitudinal gradient studied (Table 3). The abundance of species in the three zones should be studied further to determine associated factors such as seed production in the zone or the ease of propagation through natural regeneration, as reported by Villanueva et al. [50] and Esquivel [51] in cattle ranches in Costa Rica.

*3.3. Tree Density, Conservation Status in Ecuador and IUCN*

Analysis of tree species density in the different altitudinal zones studied reveals notable patterns of distribution and abundance. In the lower zone (400–700 masl), *Jacaranda copaia* leads with the highest density, averaging 84 individuals per hectare (ind/ha). It is followed by *Psidium guajava*, with a density of 48 ind/ha, and *Cordia alliodora* with 18 ind/ha. In contrast, in the middle zone (701–1600 masl), *Piptocoma discolor* stands out as the dominant species, with an average density of 26 ind/ha. It is followed by *Cordia alliodora* with 18 ind/ha and *Cedrela odorata* with 12 ind/ha. These findings show, on the one hand, the knowledge of the timber markets on the part of the producers in these areas where, according to Torres et al. [28] and Mejía et al. [52], wood is still harvested as part of their livelihoods, both formally and informally [53]. Finally, in the high zone (1601–2000 masl), *Sterculia tessmannii* emerges as the dominant species, with an average density of 37 ind/ha, while *Psidium guajava* has 28 ind/ha and *Inga* sp. reports 16 ind/ha (Table 4). Regarding the species identified, their use in silvopastoral systems in the Ecuadorian Amazon is frequent [7] due to their high commercial value in terms of timber quality, such as *Jacaranda copaia*, *Cordia alliodora*, *Cedrela odorata*, *Sterculia tessmannii*, *Cordia alliodora* and

*Cedrela odorata* [54–56], in addition to providing shade for livestock [57] and storing atmospheric carbon. Meanwhile, *Psidium guajava* and *Inga* ssp. stand out for their importance in the production of fruits for both human and livestock consumption [7,58,59]. In addition, several authors [60,61] suggest that these species contribute to the improvement of soil quality by means of nitrogen fixation, as in the case of *Inga* sp.

**Table 4.** Tree species density, status in Ecuador with their IUCN conservation status and wood densities for tree species in silvopastoral systems in low zone (N = 12), middle zone (N = 8) and *high zone* (N = 6) in total of 26 plots (2826 m$^2$) along the studied gradient in Napo, SBR.

| Species | Tree Density (Ind $\geq$ 10 cm DAP/ha) | | | Status | * Category IUCN | Wood Density g/cm$^3$ |
|---|---|---|---|---|---|---|
| | **Low** | **Middle** | **High** | | | |
| *Annona papilionella* | 0 | 1 | 0 | Native | LC | 0.48 |
| *Annona* sp. | 9 | 1 | 0 | Native | NE | 0.47 |
| *Rollinia* sp. | 0 | 0 | 1 | Native | NE | 0.61 |
| *Apeiba membranaceae* | 1 | 0 | 0 | Native | NE | 0.27 |
| *Bactris gasipaes* | 9 | 0 | 0 | Native | NE | 0.43 |
| *Iriartea deltoidea* | 3 | 1 | 0 | Native | LC | 0.27 |
| *Wettinia maynensis* | 0 | 5 | 0 | Native | LC | 0.31 |
| *Piptocoma discolor* | 6 | 26 | 0 | Native | LC | 0.47 |
| *Vernonanthura patens* | 2 | 0 | 0 | Native | LC | 0.54 |
| *Crescentia cujete* | 1 | 0 | 0 | Native | LC | 0.70 |
| *Jacaranda copaia* | 84 | 3 | 0 | Native | LC | 0.60 |
| *Cordia alliodora* | 0 | 18 | 0 | Native | LC | 0.51 |
| *Dacryodes peruviana* | 0 | 6 | 0 | Native | LC | 0.61 |
| *Protium nodulosum* | 0 | 4 | 0 | Native | LC | 0.55 |
| *Calophyllum brasiliense* | 1 | 1 | 0 | Native | LC | 0.47 |
| *Terminalia oblonga* | 1 | 0 | 0 | Native | LC | 0.69 |
| *Cordia alliodora* | 122 | 0 | 0 | Native | LC | 0.51 |
| *Croton lechleri* | 0 | 0 | 1 | Native | NE | 0.47 |
| *Sapium glandulosum* | 0 | 1 | 0 | Native | LC | 0.44 |
| *Dussia tessmannii* | 2 | 0 | 0 | Native | LC | 0.47 |
| *Erythrina poeppigiana* | 1 | 0 | 0 | Native | LC | 0.47 |
| *Inga edulis* | 2 | 6 | 0 | Native | LC | 0.51 |
| *Inga* sp. | 0 | 8 | 16 | Native | NE | 0.57 |
| *Piptadenia pteroclada* | 9 | 0 | 0 | Native | LC | 0.76 |
| *Nectandra* sp. | 14 | 3 | 3 | Native | NE | 0.53 |
| *Ocotea* sp. | 12 | 0 | 14 | Native | NE | 0.54 |
| *Persea americana* | 1 | 0 | 0 | Native | LC | 0.60 |
| *Grias neuberthii* | 8 | 3 | 0 | Native | LC | 0.62 |
| *Ceiba samauma* | 2 | 0 | 0 | Native | NE | 0.57 |
| *Heliocarpus americanus* | 0 | 0 | 0 | Native | LC | 0.47 |
| *Sterculia tessmannii* | 2 | 2 | 37 | Native | LC | 0.47 |
| *Miconia* sp. | 7 | 0 | 0 | Native | NE | 0.63 |
| *Cabralea canjerana* | 5 | 0 | 0 | Native | LC | 0.53 |
| *Cedrela montana* | 0 | 0 | 1 | Native | VU | 0.47 |
| *Cedrela odorata* | 16 | 12 | 1 | Native | VU | 0.44 |
| *Brosimum guianense* | 2 | 0 | 0 | Native | LC | 0.47 |
| *Ficus cuatrecasana* | 0 | 0 | 1 | Native | NE | 0.47 |
| *Ficus maxima* | 2 | 1 | 0 | Native | LC | 0.47 |
| *Ficus* sp. | 6 | 0 | 10 | Native | NE | 0.42 |
| *Virola flexuosa* | 16 | 0 | 0 | Native | LC | 0.47 |
| *Psidium guajava* | 48 | 5 | 28 | Native | LC | 0.71 |
| *Hieronyma oblonga* | 0 | 0 | 1 | Native | LC | 0.47 |
| *Citrus limon* | 1 | 1 | 2 | No Native | NE | 0.71 |
| *Citrus sinensis* | 7 | 0 | 0 | No Native | NE | 0.71 |
| *Pouteria caimito* | 6 | 2 | 1 | Native | LC | 0.81 |
| *Pouteria* sp. | 0 | 0 | 0 | Native | NE | 0.77 |
| *Cecropia membranacea* | 0 | 3 | 0 | Native | LC | 0.33 |
| *Cecropia* sp. | 1 | 0 | 0 | Native | NE | 0.36 |
| *Pourouma cecropiifolia* | 4 | 1 | 0 | Native | LC | 0.36 |
| *Vochysia braceliniae* | 28 | 0 | 0 | Native | LC | 0.39 |
| *Vochysia ferruginea* | 6 | 0 | 0 | Native | LC | 0.36 |

* Categories IUCN: LC = least concern, NE = not evaluated, VU = vulnerable.

Regarding biological distribution, it was observed that of the 51 taxa analyzed in the three different altitudinal zones, 49 were native and two were non-native: *Citrus limon* and *Citrus sinensis* (Rutaceae). In relation to the number of native species, this may be related to the fact that the cattle ranchers of the SRB still conserve up to 40% of the area of forest around the pastures on their farms [10]. Meanwhile, as far as non-native species are concerned, the use of *C. limon* and *C. sinensis* in pastures is frequent due to their benefits as a source of dietary supplementation for dairy cattle for their antioxidant, antimicrobial, antistress and anti-inflammatory properties [62,63].

From the 51 taxa analyzed, according to the IUCN conservation categories, the findings indicate that a total of 33 species (64.7%) could currently be classified as "Least Concern" (LC) species. In addition, two species (3.9%) were identified as "Vulnerable" (Vu): *Cedrela odorata* (in the three zones) and *Cedrela montana* (in the high zone). It was an important finding given that in the Ecuadorian Amazon, seven species of genus *Cedrela* have been reported in primary forests [64,65] from which two are found in pastures with dispersed trees, contributing to their conservation.

Regarding the percentage of trees classified as LC species, this is similar to that reported by López-Tobar et al. [55] who analyzed 214 of the most commercialized timber species in the Amazon and found that 67.6% (142 sp.) are currently classified as LC species. Likewise, 16 taxa (31.4%) were recorded as being in the "Not Evaluated" (NE) category according to the latest IUCN update. Of these, seven have been taxonomically identified down to the species: *Apeiba membranaceae*, *Bactris gasipaes*, *Croton lechleri*, *Ceiba samauma*, *Ficus cuatrecasana*, *Citrus limon* and *Citrus sinensis*. This suggests the need for further research efforts to catalog unassessed species, which according to our results corresponds to 31.4%.

In addition, nine of the taxa are described at the genus level: *Annona* sp., *Rollinia* sp., *Inga* sp., *Nectandra* sp., *Ocotea* sp., *Miconia* sp. and *Ficus* sp. This pattern of nonevaluated species is similar to that reported by Guevara et al. [66] who in their findings highlight that 89% of the lowland tree species in the RAE have not been evaluated by the IUCN. Similarly, López-Tobar et al. [55] suggest that 28% (60 sp.) of the most traded timber species in the last 10 years (2012–2021) do not have a current conservation category according to the latest IUCN red list update.

In addition, the wood density for the 51 taxa reported in the three altitudinal zones of the SBR is presented in Table 4. In general terms, the species with the highest density was *Pouteria caimito* with an average density of 0.81 g/cm$^3$. In addition, species with notable density were identified as *Piptadenia pteroclada* with 0.76 g/cm$^3$, followed by *Psidium guajava*, *Citrus limon* and *Citrus sinensis*, all with the same average density of 0.71 g/cm$^3$. On the other hand, the presence of these species in pastures may be related to the fact that their weight and hardness make them highly desirable in the production of materials used for the manufacture of doors and floors in the construction and fine cabinet-making industries [56].

*3.4. Main Reported Uses of Tree and Palm Species*

Table 5 provides a detailed analysis of 48 forest species and their applications in categories such as food, medicine, handicrafts and construction. Among these species, 20 were identified as useful for human and livestock food, with *Psidium guajava* (Myrtaceae) and *Pouteria caimito* (Sapotaceae) highlighted for their nutritional value [67]. In the medicinal field, 22 species were recognized for their medicinal properties, *Citrus limon* (Rutaceae) and *Jacaranda copaia* (Bignoniaceae) being particularly noteworthy [68,69]. For artisanal use, 17 species were recorded, with *Erythrina poeppigiana* (Fabaceae) and *Cedrela odorata* (Meliaceae) standing out for their versatility and demand [44]. Finally, in the construction material category, 25 species were identified, including *Terminalia oblonga* (Combretaceae) and *Vochysia ferruginea* (Vochysiaceae), known for the quality of their wood [70]. It is notable that several species, such as *Annona papilionella* (Annonaceae), *Apeiba membranacea* (Arecaceae) and *Dacryodes peruviana* (Burseraceae), stand out for their versatility, being used in all the categories evaluated [44,71,72]. This multifunctionality of forest species empha-

sizes their intrinsic value and the need to adopt conservation and sustainable management strategies [73,74].

**Table 5.** Use of plants reported in trees, palms and fruit trees in silvopastoral systems in the low zone (N = 12), middle zone (N = 8) and high zone (N = 6) in total of 26 plots (2826 m$^2$) along the studied gradient in Napo, SBR.

| Scientific Name | Family | Use | | | |
|---|---|---|---|---|---|
| | | Human and Livestock Feed | Medicinal Use | Craft Use | Material for Construction |
| *Annona papilionella* | Annonaceae | √ | √ | √ | √ |
| *Annona* sp. | | √ | √ | √ | √ |
| *Rollinia* sp. | | √ | √ | √ | √ |
| *Apeiba membranacea* | Arecaceae | √ | √ | √ | √ |
| *Bactris gasipaes* | | √ | √ | √ | √ |
| *Iriartea deltoidea* | | √ | √ | √ | √ |
| *Wettinia maynensis* | | √ | √ | √ | √ |
| *Piptocoma discolor* | Asteraceae | √ | √ | | √ |
| *Vernonanthura patens* | | | √ | √ | √ |
| *Crescentia cujete* | Bignoniaceae | √ | √ | | √ |
| *Jacaranda copaia* | | | √ | | √ |
| *Cordia alliodora* | Cordiaceae | | √ | | √ |
| *Dacryodes peruviana* | Burseraceae | √ | √ | √ | √ |
| *Protium nodulosum* | | √ | √ | √ | √ |
| *Calophyllum brasiliense* | Calophyllaceae | | √ | | √ |
| *Terminalia oblonga* | Combretaceae | | | √ | √ |
| *Croton lechleri* | Euphorbiaceae | | √ | | √ |
| *Sapium glandulosum* | | √ | √ | | √ |
| *Dussia tessmannii* | Fabaceae | | | √ | √ |
| *Erythrina poeppigiana* | | | | | √ |
| *Inga edulis* | | √ | √ | | √ |
| *Inga* sp. | | √ | √ | | √ |
| *Piptadenia pteroclada* | | | √ | | √ |
| *Nectandra* sp. | Lauraceae | | | | √ |
| *Ocotea* sp. | | | √ | | √ |
| *Persea americana* | | √ | √ | | √ |
| *Grias neuberthii* | Lecythidaceae | √ | √ | | √ |
| *Ceiba samauma* | Malvaceae | | | √ | √ |
| *Heliocarpus americanus* | | | √ | √ | √ |
| *Sterculia tessmannii* | | | | √ | √ |
| *Miconia* sp. | Melastomataceae | | √ | | √ |
| *Cabralea canjerana* | Meliaceae | | | | √ |
| *Cedrela montana* | | | | | √ |
| *Cedrela odorata* | | | √ | | √ |
| *Brosimum guianense* | Moraceae | √ | | | √ |
| *Ficus cuatrecasana* | | | | | √ |
| *Ficus maxima* | | √ | √ | | √ |
| *Ficus* sp. | | √ | √ | | √ |
| *Virola flexuosa* | Myristicaceae | | | | √ |
| *Psidium guajava* | Myrtaceae | √ | √ | | √ |
| *Hieronyma oblonga* | Phyllanthaceae | √ | | | √ |

**Table 5.** *Cont.*

| Scientific Name | Family | Use | | | |
|---|---|---|---|---|---|
| | | Human and Livestock Feed | Medicinal Use | Craft Use | Material for Construction |
| *Citrus limon* | Rutaceae | √ | √ | | √ |
| *Citrus sinensis* | | √ | √ | | √ |
| *Pouteria caimito* | Sapotaceae | √ | √ | | √ |
| *Pouteria* sp. | | √ | √ | | √ |
| *Cecropia membranacea* | Urticaceae | √ | √ | | √ |
| *Cecropia* sp. | | √ | √ | | √ |
| *Pourouma cecropiifolia* | | √ | | | √ |
| *Vochysia braceliniae* | Vochysiaceae | | √ | | √ |
| *Vochysia ferruginea* | | | | √ | √ |

Source: De la Torre et al. [44].

### 3.5. Contributions to Landscape Conservation Planning

In November 2016, the Ministry of Environment of Ecuador (MAE) enacted the REDD+ Action Plan "Forests for Good Living", which seeks to articulate the measures and actions inside and outside the forest, including national and local policies, programs and initiatives, as well as generate multiple environmental and social benefits; one of the objectives of the plan is to "support the transition towards sustainable productive systems free from deforestation-free production systems" [75].

In this context of planning for conservation, our findings highlight the role of silvopastoral systems in landscape conservation, particularly in biodiversity-rich areas like the SBR. These systems, contrasting with conventional cattle ranching, significantly enhance the conservation of native species due to their high diversity of shade trees [76]. Our study identified key species with conservation concerns, such as *Cedrela odorata* and *Cedrela montana*, listed as VU on the IUCN Red List (Table 4), underscoring the critical role of silvopastoral landscapes in maintaining species under threat. However, it is noteworthy that the conservation status of 31.4% of the documented tree species remains unassessed, indicating a gap in our understanding of their ecological significance.

The transformation of the SBR region's vegetation, predominantly due to deforestation [77,78], emphasizes the need for integrated landscape management approaches. Trees in pasturelands are fundamental for preserving native tree species and their genetic diversity and also serve as refuges for disturbance-tolerant species [79]. This study underlines the necessity of incorporating silvopastoral systems into EAR conservation planning, recognizing their value not only in agricultural productivity but also in safeguarding vulnerable tree species and maintaining essential genetic diversity in threatened ecosystems [80–82].

The considerable diversity of tree species in these grasslands with dispersed trees has a direct and quantifiable impact on carbon sequestration [13], a critical ecosystem service in the context of global climate change mitigation. This relationship between tree diversity and carbon storage capacity offers a tangible pathway for the development of environmental incentives for local producers [83]. As Torres et al. [84] suggest, these incentives could be strategically directed to reinforce best management practices (BMPs) that not only bolster carbon sequestration but also promote overall ecosystem health. But such implementation should strongly consider the livelihoods of producers and the opportunity cost of livestock activities along the altitudinal gradient [85], issues that have already been explored in this area. Therefore, this evidence on the ecological importance value of dispersed trees in grasslands would complement the other studies carried out in this landscape. Therefore, implementing BMPs in this context not only aligns with ecological sustainability goals but also provides a framework for local producers to contribute actively to climate change mitigation efforts [75,86]. The potential for such practices to generate verifiable carbon credits could further integrate environmental services into the economic mainstream of local

communities, thereby creating a synergistic relationship between conservation initiatives and economic development [4,87].

## 4. Conclusions

The tree characterization across altitudinal zones showed significant differences. Factors such as zone use, age, production system and pasture management influence tree diameter distribution, density and floristic composition. The low and middle zones of the altitudinal gradient showed a greater number of individuals and floristic richness in the silvopastoral systems; this positive relationship was associated with a positive regeneration of tree species and their abundance, given that in these zones, there is a greater amount of remnant forest surrounding the pastures. In these areas, species of high commercial value such as *Cedrela odorata*, *Jacaranda copaia*, *Piptocoma discolor* and *Sterculia tessmannii* predominate, which indicates the knowledge of the timber markets by the producers in these areas.

The use of the IVI helped to identify that only the 10 most important tree species in the pastures with trees dispersed along the altitudinal gradient studied represented approximately more than 70% of the IVI in the low and middle zones and up to 96% in the high zone. This suggests further studies should focus on the factors associated with the abundance of these species, as well as the design of strategies for greater species diversification in pastures, as measures to promote valuable species in an area of high diversity and endemism such as the SBR.

In this study, a total of 51 tree species were recorded, of which 49 are of native origin and two are exotic species. Among the most frequent species throughout the altitudinal range were *Jacaranda copaia*, *Psidium guajava*, *Cordia alliodora*, *Piptocoma discolor* and *Sterculia tessmannii*, which are widely known for their use in terms of timber and fruit for human consumption and livestock. In addition, the presence of *Citrus limon* and *Citrus sinensis* was identified, which are frequently used due to their nutritional and biological benefits for livestock. On the other hand, it was found that *Cordia alliodora* reports the highest density recorded in the study area. Regarding the conservation categories of the IUCN, 64.7% of the species analyzed were classified as LC species, followed by 31.4% as NE and 3.9% as VU.

The results support the importance of adapting sectoral policies to different altitudes in silvopastoral systems, especially focused on the protection of vulnerable species such as *C. odorata* and *C. montana*, which are crucial for the conservation of forest biodiversity in pastures. The significant presence of species not evaluated by the IUCN, which constitute 31% of the identified forest species, reveals a vital opportunity for their inclusion in future evaluations and deepening of their knowledge. This approach is crucial not only to understand the ecology and distribution of these species in silvopastoral systems but also to identify those at risk and develop appropriate conservation strategies, thus ensuring the protection and sustainable management of both species already identified as vulnerable, particularly *Cedrela* species, as well as those that remain to be evaluated in these dynamic and essential ecosystems.

**Author Contributions:** Conceptualization, B.T. and A.G.; methodology, B.T. and A.T.-N.; software, B.T.; validation, R.J.H.-F., C.B. and A.T.-N.; formal analysis, B.T.; investigation, B.T., C.B. and A.T.-N.; data curation B.T. and R.J.H.-F.; writing—original draft preparation, B.T., R.J.H.-F., C.B., A.T.-N. and A.G.; writing—review and editing, A.G., B.T., A.T.-N. and R.J.H.-F.; supervision B.T. and A.G. All authors have been involved in developing, writing, commenting, editing and reviewing the manuscript. All authors have read and agreed to the published version of the manuscript.

**Funding:** This research received no external funding.

**Institutional Review Board Statement:** The study was conducted in accordance with the Declaration of Helsinki and was approved by the Ethics Committee of Pontificia Universidad Católica del Ecuador Sede Ibarra on 4 April 2019.

**Informed Consent Statement:** Informed consent was obtained from all subjects involved in the study.

**Data Availability Statement:** Data are available from the corresponding authors upon request.

**Acknowledgments:** This work is part of the results of a joint research agreement between the Amazon State University (UEA) and Rainforest Alliance Inc. The authors thank Deutsche Gesellschaft für Internationale Zusammenarbeit (GIZ), through the REDD + Early Movers Program (REM) and Rainforest Alliance Inc., for all the support. We also thank the MAG, MAATE, UEA, UTEQ and ECONGEST AGR267 Group at Cordoba University for their support during the fieldwork stage, as well as the households in the three zones that shared valuable information about their livestock activities.

**Conflicts of Interest:** The authors declare no conflicts of interest.

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
