# Peer review of "Tree Diversity and Its Ecological Importance Value in Silvopastoral Systems: A Study along Elevational Gradients in the Sumaco Biosphere Reserve, Ecuadorian Amazon"

_land, doi:10.3390/land13030281_

Round 1
Reviewer 1 Report
Comments and Suggestions for Authors
The topic of research carried out by the authors is very relevant cosidering the importance of trees in silvopastoral systems towards varieties of ecosystem services including climate change mitigation.The manuscript is written well and deserves appreciation.However manuscript needs improvement in structure before it is finally accepted.Authors are requested to respond to the queries as mentioned below:
1. Absreact: line 30: Cedrela montana (not Montana)
2. Data analysis:
(i) line 112-113: Please make necessary corrections calculating Relative abundance and Relative Dominance.
(ii) line 102:please mention which parameter was used for calculating Shannon's diversity index and provide formula for the same.
3.Results and Discussion:
(i) line 158-159 Please make necessary modifications in table 2: Maximum DBH in column 5, please check the value 41.54 which seems incorrect.It would be better to give minimum and maximum DBH values instead of only maximum DBH values for low, middle and high elevation zones.
(ii) line 255-264-please indicate the purpose of mentioning wood density values in table 4.Please mention the source for the wood density values in methodology section.
(iii) Please discuss the importance of legume trees in silvopastoral systems, if any.
Author Response
Dear reviewer:
On behalf of all the authors of the article entitled: Tree diversity and its ecological importance value in silvopastoral systems: A study along elevational gradients in the Sumaco Biosphere Reserve, Ecuadorian Amazon; I appreciate your kind comments and suggestions, as they have allowed us to improve the scientific quality of the manuscript. Below we present in detail and by number of lines the changes made to the text:
Reviewer 1: The topic of research carried out by the authors is very relevant considering the importance of trees in silvopastoral systems towards varieties of ecosystem services including climate change mitigation. The manuscript is written well and deserves appreciation. However manuscript needs improvement in structure before it is finally accepted. Authors are requested to respond to the queries as mentioned below:
Reviewer 1: Absreact: line 30: Cedrela montana (not Montana)
Dear Reviewer
Thank you very much for your valuable suggestions.
We have thoroughly checked all scientific names and have inserted them correctly.
Reviewer 1: line 112-113: Please make necessary corrections calculating Relative abundance and Relative Dominance.
This part of the paper It has been modified. In the text the formula for calculating relative abundance and dominance has been corrected.
Reviewer 1: line 102: please mention which parameter was used for calculating Shannon's diversity index and provide formula for the same. Authors:
We have incorporated in the text, the parameters used to determine the Shannon diversity index and the formula used.
Reviewer 1: line 158-159 Please make necessary modifications in table 2: Maximum DBH in column 5, please check the value 41.54 which seems incorrect. It would be better to give minimum and maximum DBH values instead of only maximum DBH values for low, middle and high elevation zones.
Thanks for the comment, indeed we have checked the maximum DBH value which is correct, since it is the maximum DBH measurement found, which is 41.54 cm on the high elevation and consequently the same value in the whole study area. Regarding the minimum diameter we consider that it is not necessary, because the inventory was only considered from 10 cm DBH in all the plots and 10 cm DBH is the minimum value.
Reviewer 1: line 255-264-please indicate the purpose of mentioning wood density values in table 4. Please mention the source for the wood density values in methodology section
We have drafted a paragraph in the methodology mentioning the importance of including density values, using the Global Wood Density Database (Chave et al., 2009; Zanne et al., 2009):
Zanne AE, Lopez-Gonzalez G, Coomes DA, Ilic J, Jansen S, Lewis SL, Miller RB, Swenson NG, Wiemann MC, Chave J (2009) Data from: Towards a worldwide wood economics spectrum. Dryad Digital Repository. https://doi.org/10.5061/dryad.234
Chave J, Coomes DA, Jansen S, Lewis SL, Swenson NG, Zanne AE (2009) Towards a worldwide wood economics spectrum. Ecology Letters 12(4):351-366
Reviewer 1: Please discuss the importance of legume trees in silvopastoral systems, if any.
Regarding your suggestion, we have discussed the importance of these species in silvopastoral systems in previous research. That is why, in our research, Torres et al. [1] mentioned that legumes are often used to increase the protein and mineral content of calcium (Ca) and phosphorus (P), improve animal nutrition and fix nitrogen in the soil through symbiosis with bacteria of the genus Rhizobium.
In this research, we suggest agro-ecological management alternatives in agroforestry systems, Such as, protein banks, combination of grasses and legumes, and the incorporation of indigenous breeds, among other. All these good agricultural and livestock practices, are recommended for the entire EAR, including the study area.
References
- Torres, B.; Andrade, V.; Heredia-R, M.; Toulkeridis, T.; Estupiñán, K.; Luna, M.; Bravo, C.; García, A. Productive Livestock Characterization and Recommendations for Good Practices Focused on the Achievement of the SDGs in the Ecuadorian Amazon. Sustainability 2022, 14, 10738.

Reviewer 2 Report
Comments and Suggestions for Authors
Comments:
1. In Figure 1, I am not sure whether the numbers around the picture represent latitude and longitude. If so, please change it to the latitude and longitude format we are familiar with. If not, please add the latitude and longitude information of the study area.
2. In Table 5, please do not use the "×" symbol to represent that the tree species have been utilized. This is very easy to cause misunderstanding. I recommend using other symbols that are less likely to cause ambiguity, such as "√". If you insist on using "×", you need to mark it below the table.
3. The article describes the selection criteria for the 26 temporary plots in 2.2, so on top of this, what is the basis for the layout of these 26 temporary plots? Is it laid out along elevation sections or randomly selected? Please describe it.
4. In 2.3, the article uses basal area to calculate relative dominance. Since the diameter at breast height has been measured in the survey, why not use the more convincing cross-sectional area of breast height to calculate relative dominance?
5. The title of 3.1 is "Richness and floristic composition of silvopastoral system", but I did not see any content about species composition in this section. Are there any adjustments to the title or content that should be considered?
6. In the previous paragraph of Table 2, the article writes "maximum DBH did not show significant differences", but it can be seen from Table 2 that the difference in maximum DBH is quite significant. Please ensure the accuracy of the description.
7. The reasons for the higher density of some tree species were explained in 3.3, but I think we can think more from other perspectives, such as what role do pasture animals play in it? What effect do they have on trees? Do tree seedlings with high density have the characteristics of resistance to trampling and lodging?
8. The IUCN ratings in Table 4 have been updated, and the ratings of some tree species have changed. Please update in time. Also note the distinction between "DD" and "NE".
9. Regarding the differences in tree richness at different altitudes, the article focuses on describing the role of human factors in it. It does not devote much ink to other ecological factors such as climate and topography. I think it can be discussed appropriately. . In addition, the impact of habitat changes caused by altitude changes on tree species is also lacking.
Comments on the Quality of English LanguageIn previous editions, I saw quite a few Spanish words in the article. And in the latest version I'm glad you changed them into English. I hope you will carefully check your grammar again to make it comply with English writing standards.
Author Response
Dear reviewer 2:
On behalf of all the authors of the article entitled: Tree diversity and its ecological importance value in silvopastoral systems: A study along elevational gradients in the Sumaco Biosphere Reserve, Ecuadorian Amazon; I appreciate your kind comments and suggestions, as they have allowed us to improve the scientific quality of the manuscript. Below we present in detail and by number of lines the changes made to the text:
Reviewer 2: In Figure 1, I am not sure whether the numbers around the picture represent latitude and longitude. If so, please change it to the latitude and longitude format we are familiar with. If not, please add the latitude and longitude information of the study area.
Authors:
Thank you very much for your valuable suggestions. We have made the suggested changes to the map.
Reviewer 2: In Table 5, please do not use the "×" symbol to represent that the tree species have been utilized. This is very easy to cause misunderstanding. I recommend using other symbols that are less likely to cause ambiguity, such as "√". If you insist on using "×", you need to mark it below the table
Authors:
We have modified what you requested; you can check it in table 5.
Reviewer 2: The article describes the selection criteria for the 26 temporary plots in 2.2, so on top of this, what is the basis for the layout of these 26 temporary plots? Is it laid out along elevation sections or randomly selected? Please describe it.
Authors:
The sample size used in previous research in primary forests was 5 plots of 1000 m2 in each altitudinal gradient, with results already published in Torres et al., (2019). However, in this case and in the same area, 12, 8 and 6 plots were obtained in the low, middle and high zones, respectively, each temporary plot with an area of 2,826 m2. The difference in the number of plots was due to the willingness of the producers to participate in this research.
Additionally, we have incorporated in the table the number of plots and the area sampled (ha) in each altitudinal gradient.
Reviewer 2: In 2.3, the article uses basal area to calculate relative dominance. Since the diameter at breast height has been measured in the survey, why not use the more convincing cross-sectional area of breast height to calculate relative dominance?
Authors:
In fact, the cross-sectional area of breast height to calculate relative dominance has been measured, probably we have not expressed ourselves well. We have clarified it.
Reviewer 2: The title of 3.1 is "Richness and floristic composition of silvopastoral system", but I did not see any content about species composition in this section. Are there any adjustments to the title or content that should be considered?
Authors:
Done, we have changed the title to wealth, diversity index and structural parameters.
Reviewer 2: In the previous paragraph of Table 2, the article writes "maximum DBH did not show significant differences", but it can be seen from Table 2 that the difference in maximum DBH is quite significant. Please ensure the accuracy of the description.
Authors:
The maximum value is only the maximum value of DBH size found in each altitudinal gradient, which is reported only to indicate the diameter characteristic of each zone, but does not correspond to average values.
Reviewer 2: The reasons for the higher density of some tree species were explained in 3.3, but I think we can think more from other perspectives, such as what role do pasture animals play in it? What effect do they have on trees? Do tree seedlings with high density have the characteristics of resistance to trampling and lodging?
Authors:
For one side, animals have two effects on tree growth, one is foraging, and the other is walking. With the objective to avoid these direct effects, we have only measured trees with a DBH greater than 10 cm DBH, considering that in the study area, we have only measured trees with a DBH greater than 10 cm DBH. Besides that, the trees in these pastures are not planted, they are mostly relic trees that have remained after changing the use of the land and others of natural regeneration which the producers take care, when they know that they are trees with timber potential for sale or generators of fodder for livestock.
Reviewer 2: The IUCN ratings in Table 4 have been updated, and the ratings of some tree species have changed. Please update in time. Also note the distinction between "DD" and "NE".
I hope this explanation clarifies your concerns.
The category "Data Deficient" (DD) is assigned to species for which some information is available, but not enough to assess their conservation status. On the other hand, "Not Evaluated" (NE) refers to species that have not been assessed at all by the organization or authority that classifies conservation status. In short, 'DD' indicates some but insufficient information, while 'NE' indicates that no assessment has been made.
We have also included a short explanation of each acronym at the bottom of the table.
Reviewer 2: Regarding the differences in tree richness at different altitudes, the article focuses on describing the role of human factors in it. It does not devote much ink to other ecological factors such as climate and topography. I think it can be discussed appropriately. . In addition, the impact of habitat changes caused by altitude changes on tree species is also lacking.
The classical literature talks mostly about ecological, climatic or altitudinal factors determining richness, especially in natural environments.
In the case of natural environments with anthropogenic intervention, it is important to focus on man because he models the system according to his knowledge and needs. This knowledge contributes to designing appropriate public policies to promote sustainable livestock systems.
some of these issues we have placed in the discussion like that:
Section 3.1: This pattern is contrary to the results of species richness found in primary forests in the same area of SBR by Torres et al. (2020) in the Ecuadorian Amazon, as well as in a protected primary forest in the Tumbesian dry forest ecoregion (Graefe et al., 2020) where the number of tree species increased with increasing altitude. This variation trend in primary forests may be due to the fact that species distribution patterns are the result of multiple ecological processes [41], influenced by geographic differences and environmental factors such as climate and soil [42]. But on the other hand, a decreasing species richness with increasing altitude in anthropogenic systems such as the dispersed trees in these pasture systems could be the results of different factors. Anthropogenic factors determined the establishment and management of these production systems generated by populations of mestizo settlers who came at different times, cultural backgrounds and inheritance of the ecosystem.

Reviewer 3 Report
Comments and Suggestions for Authors
The manuscript titled: "Tree diversity and its ecological importance value in silvopas-toral systems: A study along elevational gradients in the Su-maco Biosphere Reserve, Ecuadorian Amazon" presents research results from the Amazon area, which is one of the world's centers of biological diversity. The authors conducted research on trees in a silvo-pastoral landscape. For research, they selected 26 plots located in three height zones above sea level. The research results are interesting and deserve to be published. Most of the data was properly presented and discussed. However, I have some doubts about the material and research methods. This chapter needs to be completed. The results and conclusions also require refinement. Below are my comments.
Detailed comments by line numbers:
21-23 – How did the authors determine the influence of settlement history and pasture management on differences in tree diameter structure?
28 – “51 tree species” – If there are taxa of different taxonomic rank, they cannot all be classified into species. The authors should write that they found 51 taxa (including 42 species and 9 taxa at the rank of genus).
76-87 – Do these three zones have the same type of vegetation? If there are differences in the vertical distribution of vegetation (plant zones), the authors should write about it and explain why they compared three different plant and climate zones.
92-95 – How many plots were there in each zone? The three altitude zones can differ significantly from each other. Table 1 contains only a few variables. It is not known what the settlement and management looked like in recent years. Was it similar in all zones? E.g.: how often was grazing (how long did it last)? What species of animals were grazed? How many cattle were there per 1 ha of pasture? The authors should also characterize the environmental conditions in more detail, e.g. terrain slope, exposure, soil type.
136-142 – And what about this study? The authors should explain what causes this difference.
164-165 – In Figure 3, there are no captions "a), b)" on the charts.
166-176 – How to explain it? There is no comment on the reasons for the differences in the distribution of trunk diameter at the three height levels.
190-197 – Why do certain species dominate in a given altitude zone? Is it due to the biology and ecology of the species, history of use, terrain...?
201-203 – Why aren't all 51 taxa listed in Table 3?
225 – “51 species” – change to 55 taxa
301-303 – Why didn't the authors propose protective categories for these species?
330-332 – First, you need to characterize the factors (use, production system, management) and then you can talk about their impact on the distribution of tree diameter, density and floristic composition (see: notes to the material and methods chapter).
356-359 – And what are these policies? How can we know whether sectoral policies are aligned if this is not presented in the manuscript?
Technical notes
There is no need to specify Family in every table.
Tables should be formatted according to one scheme.
Author Response
Dear reviewer:
On behalf of all the authors of the article entitled: Tree diversity and its ecological importance value in silvopastoral systems: A study along elevational gradients in the Sumaco Biosphere Reserve, Ecuadorian Amazon; I appreciate your kind comments and suggestions, as they have allowed us to improve the scientific quality of the manuscript. Below we present in detail and by number of lines the changes made to the text:
Reviewer 3: The manuscript titled: "Tree diversity and its ecological importance value in silvopastoral systems: A study along elevational gradients in the Sumaco Biosphere Reserve, Ecuadorian Amazon" presents research results from the Amazon area, which is one of the world's centers of biological diversity. The authors conducted research on trees in a silvopastoral landscape. For research, they selected 26 plots located in three height zones above sea level. The research results are interesting and deserve to be published. Most of the data was properly presented and discussed. However, I have some doubts about the material and research methods. This chapter needs to be completed. The results and conclusions also require refinement. Below are my comments.
Reviewer 3: 21-23 – How did the authors determine the influence of settlement history and pasture management on differences in tree diameter structure?
Dear Reviewer.
Thank you very much for the comment, we have removed this sentence, and improved the sense of the statement in the text.
Reviewer 3: 28 – “51 tree species” – If there are taxa of different taxonomic rank, they cannot all be classified into species. The authors should write that they found 51 taxa (including 42 species and 9 taxa at the rank of genus).
Authors:
We have modified this comment both in the executive summary and throughout the document.
Reviewer 3: 76-87 – Do these three zones have the same type of vegetation? If there are differences in the vertical distribution of vegetation (plant zones), the authors should write about it and explain why they compared three different plant and climate zones.
The study area SBR is rich in diversity, although the three zones are climatically different in altitude and temperature, they are part of the Andean-Amazon vegetation. In fact, our objective is to study the richness, abundance of trees and their IUCN conservation status of the silvopastoral systems along the elevational gradients.
Reviewer 3: 92-95 – How many plots were there in each zone? The three altitude zones can differ significantly from each other. Table 1 contains only a few variables. It is not known what the settlement and management looked like in recent years. Was it similar in all zones? E.g.: how often was grazing (how long did it last)? What species of animals were grazed? How many cattle were there per 1 ha of pasture? The authors should also characterize the environmental conditions in more detail, e.g. terrain slope, exposure, soil type.
All the information requested in this question are very important, however we have not placed them in this paper because they are part of other recently published contributions and cited en this contribution such as:
“Productive Livestock Characterization and Recommendations for Good Practices Focused on the Achievement of the SDGs in the Ecuadorian Amazon” (Torres et al., 2022). and “Identification and Assessment of Livestock Best Management Practices (BMPs) Using the REDD+ Approach in the Ecuadorian Amazon” (Torres et al., 2021)
Reviewer 3: 136-142 – And what about this study? The authors should explain what causes this difference.
Authors:
We have improved the results and discussion and add two importance references in the following manners:
This pattern is contrary to the results of species richness found in primary forests in the same area of SBR by Torres et al. (2020) in the Ecuadorian Amazon, as well as in a protected primary forest in the Tumbesian dry forest ecoregion (Graefe et al., 2020) where the number of tree species increased with increasing altitude. This variation trend in primary forests may be due to the fact that species distribution patterns are the result of multiple ecological processes [41], influenced by geographic differences and environmental factors such as climate and soil [42]. But on the other hand, the evidence of decreasing species richness with increasing altitude in anthropic systems such as the dispersed trees in these pasture systems could be the results of the establishment and management of these production systems generated by populations of mestizo settlers who came at different times and with different cultural backgrounds and different inheritance of the ecosystem, according to Torres et al. [43]
Torres, B., Vasseur, L., López, R. et al. Structure and above ground biomass along an elevation small-scale gradient: case study in an Evergreen Andean Amazon forest, Ecuador. Agroforest Syst 94, 1235–1245 (2020). https://doi.org/10.1007/s10457-018-00342-8
Graefe, S.; Rodrigo, R.; Cueva, E.; Butz, P.; Werner, F.A. & Homeier, J. (2020): Impact of disturbance on forest structure and tree species composition in a tropical dry forest of South Ecuador. Ecotropica 22, 202002.
Reviewer 3: 164-165 – In Figure 3, there are no captions "a), b)" on the charts.
Authors:
Done, we have added the captions "a), b)" to the graphs.
Reviewer 3: 166-176 – How to explain it? There is no comment on the reasons for the differences in the distribution of trunk diameter at the three height levels.
Thanks for the interesting comment.
We have incorporated the argument in the text about: “The causes of the differences in tree abundance by diameter class are attributed to knowledge of management and years of establishment, as discussed in section 3.1.”
Reviewer 3: 190-197 – Why do certain species dominate in a given altitude zone? Is it due to the biology and ecology of the species, history of use, terrain...?
On the one hand, the grasslands in the study area are surrounded by primary forest, which contributes to their ability to self-generate, and on the other hand the anthropogenic factor makes a positive selection on a certain type of tree species, with cultural, economic and productive motivations.
The authors consider that these are the two reasons, areas with presence of these species in primary forest but also that the producers take care of these species of regeneration because they have a high commercial value, as explained in the text.
Reviewer 3: 201-203 – Why aren't all 51 taxa listed in Table 3?
Authors:
Regarding your question, the table shows the 10 species with the highest IVI (importance value index) for each altitudinal level. It is important to note that if we were to include all 51 taxa, the table would be very large. Therefore, in order to satisfy your request, we have included supplementary material listing all taxa.
Reviewer 3: 225 – “51 species” – change to 55 taxa
Authors:
We have changed "species" to "taxa".
Reviewer 3: 301-303 – Why didn't the authors propose protective categories for these species?
In Ecuador these species are regulated under a system of conditioned exploitation.
We propose the development of good practices for the conservation and management of these species in these productive systems.
Reviewer 3: 330-332 – First, you need to characterize the factors (use, production system, management) and then you can talk about their impact on the distribution of tree diameter, density and floristic composition (see: notes to the material and methods chapter).
All the information requested in this question we have not placed them in this paper because they are part of othe recently published contribution and cited en this contribution such as:
“Productive Livestock Characterization and Recommendations for Good Practices Focused on the Achievement of the SDGs in the Ecuadorian Amazon” (Torres et al., 2022).
Reviewer 3: 356-359 – And what are these policies? How can we know whether sectoral policies are aligned if this is not presented in the manuscript?
Authors:
Thanks, what you say is correct, however, currently there are no policies oriented to the development of silvopasture systems. This work contributes to the knowledge of silvopasture and tree diversity, etc. It is necessary to work on another contribution to generate policies.
Reviewer 3: Technical notes
There is no need to specify Family in every table.
Tables should be formatted according to one scheme.
Authors:
We have removed the columns family from table 4 and 5.

Reviewer 4 Report
Comments and Suggestions for Authors
Detailed comments appear in the writing. The most important general comments are the following:
Silvopathoral systems are very common today, due to human activities, since forests have been replaced by these systems. The study under consideration acquires relevance in the sense of the presence of different tree species as part of these systems and how the ecological attributes of these species change along altitude gradients. The results presented in the article are very interesting, although I suggest that the authors should make some changes, which are indicated below:
- review writing in the marked places, review scientific names in italics, how to indicate area units and avoid repeating the same words on the same line.
- How was canopy cover estimated?
- With the data in Table 1, what sense does it make to do a statistical analysis?
- I suggest using this shannon index and the Simpson index, but with the conversion proposed by Jost (2006)
- The text does not indicate how the results of the rarefaction curves were used, why not to define the percentage of completeness as well?
- check details in references section

Author Response
Dear reviewer 4:
On behalf of all the authors of the article entitled: Tree diversity and its ecological importance value in silvopastoral systems: A study along elevational gradients in the Sumaco Biosphere Reserve, Ecuadorian Amazon; I appreciate your kind comments and suggestions, as they have allowed us to improve the scientific quality of the manuscript. Below we present in detail and by number of lines the changes made to the text:
Reviewer 4: Silvopastoral systems are very common today, due to human activities, since forests have been replaced by these systems. The study under consideration acquires relevance in the sense of the presence of different tree species as part of these systems and how the ecological attributes of these species change along altitude gradients. The results presented in the article are very interesting, although I suggest that the authors should make some changes, which are indicated below:
Reviewer 4: review writing in the marked places, review scientific names in italics, how to indicate area units and avoid repeating the same words on the same line.
Authors:
Thank you very much for your valuable suggestions.
We have thoroughly reviewed each area you have marked and modified to make the manuscript perfect under your considerations.
Reviewer 4: How was canopy cover estimated?
Dear Reviewer.
Canopy cover was determined using a spherical densiometer and we had incorporated the reference (Cook et al. 1995)
Reviewer 4: With the data in Table 1, what sense does it make to do a statistical analysis?
Authors:
Thanks for the comments, following your observation, we have deleted the column from the ANOVA analysis. Instead, we have increased two rows, one to place the number of plots and the other to place the area sampled in ha.
Reviewer 4: I suggest using this shannon index and the Simpson index, but with the conversion proposed by Jost (2006)
Thank you very much for your valuable suggestions
Shannon's index (H) measures the average degree of uncertainty in the prediction of which species an individual selected at random from a set belongs to, while Simpson's index (D) is the probability that two individuals taken at random belong to the same species. Both indices are simple and similar, although Shannon's index is more common in agroforestry (Somarriba, 1999).
In our study, the Shannon index was used to measure biodiversity and later the results were evaluated using Simpson's index. The results of the Shannon index showed an index lower than 2, which shows the low biodiversity compared to a forest system. In this case, when compared to other agricultural production systems (higher than 1), it indicates greater diversity.
However, these indexes use weighted average values and are simple, which is a limitation of the study. Future work should model biodiversity more rigorously, either by parametric or non-parametric methods.
Somarriba, E. 1999. Diversidad Shannon. Agroforestería en las Américas. Vol. 6 No. 23.
Grané A. 2022. Análisis del comportamiento de índices de biodiversidad en distintos estados de estructura y complejidad de sistemas biológicos a partir de experimentos con simulación. Trabajo final para obtener el grado de Máster en Conservación de la Biodiversidad y Restauración Marino y Terrestre. Universidad de Alicante. España. 60 pp.
Reviewer 4: The text does not indicate how the results of the rarefaction curves were used, why not to define the percentage of completeness as well?
It has been incorporated that even with the results of the rarefaction curve using 6 plots in each zone, we can observe that the species richness in grasslands with trees in the high zone is much lower compared to the low and middle zones. also show the trend of the curves.
Regarding the completeness analysis, we consider it not relevant to perform this type of analysis for this study, because we took an intentional sample of at least 5 plots in each gradient. However, in this case we were able to sample a total of 26 plots: 12, 8 and 6 plots in the low, middle and high zones, respectively, each plot with an area of 2826 m2, depending on the interest of the producers in being part of the research project.
Reviewer 4: check details in references section
We have reviewed in detail the references that you have marked and have provided a solution to your request.

Round 2
Reviewer 4 Report
Comments and Suggestions for Authors
I think the article improved a little, I attached the values of equally common species for the Shannon and Simpson indices (Jost, 2006), they are more representative than the traditional indices

Author Response
Dear Editor
Following your observation, we have converted the results of the Shannon and Simpson indices using the exponential suggested by Jost (2006) to determine the diversity in both exponential Shannon index (lines 112-117) and inverse of the Simpson index (lines 132-135), incorporating these two formulas:
As well as some modification in the results section in Table 2 and (lines 198 and 200).
Thank you very much for your valuable suggestions. We believe that your suggestions help us improve the paper.
The authors
